# Bridging Immune Evasion and Vascular Dynamics for Novel Therapeutic Frontiers in Hepatocellular Carcinoma

**DOI:** 10.3390/cancers17111860

**Published:** 2025-05-31

**Authors:** Sulin Wu, Namrata Anand, Zhoubo Guo, Mingyang Li, Marcos Santiago Figueroa, Lauren Jung, Sarah Kelly, Joseph W. Franses

**Affiliations:** Section of Hematology and Oncology, Department of Internal Medicine, University of Chicago, 900 E. 57th St., KCBD 7114, Chicago, IL 60637, USA; sulin.wu@uchicagomedicine.org (S.W.); namrata.anand@bsd.uchicago.edu (N.A.); zhoubo.guo@bsd.uchicago.edu (Z.G.); milingrid@bsd.uchicago.edu (M.L.); marcos.santiagofigueroa@bsd.uchicago.edu (M.S.F.); lauren.jung@bsd.uchicago.edu (L.J.); skkelly@uchicago.edu (S.K.)

**Keywords:** hepatocellular carcinoma, tumor microenvironment, immune tolerance, angiogenesis, translational therapeutics

## Abstract

Hepatocellular carcinoma (HCC) arises within a complex milieu of interacting cells and signaling elements. Modern medical therapies for HCC target the dysfunctional immune cells and vascular endothelial cells. This review summarizes mechanistic studies and enabling technologies of the HCC tumor microenvironment that can inform subsequent biomarker and therapeutic development.

## 1. Introduction

Hepatocellular carcinoma (HCC) stands as one of the most lethal cancers worldwide, largely due to its intricate tumor biology and the profound influence of its unique tumor microenvironment (TME) [1]. Such TME changes share diverse origins in distinct chronic liver diseases—such as viral hepatitis, alcohol-related liver injury, and the increasingly prevalent metabolic-associated steatohepatitis (MASH)—leading to shared terminal carcinogenesis pathways [2]. Therefore, HCC is recognized not only for its diverse etiologies but also for the complex interplay of cellular and molecular processes that govern its progression.

Recent advances in high-resolution profiling technologies have considerably refined our understanding of HCC. Spatial omics techniques, single-cell analyses, and artificial intelligence (AI)-driven platforms are unveiling the multifaceted nature of the TME, where interactions among cancer cells, immune infiltrates, stromal components, and extracellular matrix elements orchestrate tumor growth, angiogenesis, and resistance to immune attack. These insights, along with evolving liquid biopsy strategies and multi-omics biomarker panels, are steering the field toward non-genomic precision oncology approaches.

This review summarizes the global epidemiological trends, the cellular and molecular landscape of the HCC TME, and the latest clinical advances in immunotherapy and combination treatments. We explore key aspects of HCC, from the shifting etiological factors and regional disparities in incidence and survival to the specialized roles of various cell types in the TME. Furthermore, we delve into the emerging paradigm of spatially resolved, AI-enhanced analyses that have begun to unravel the complexities of tumor heterogeneity and resistance mechanisms, setting the stage for the development of next-generation therapies. In highlighting major clinical trials, we emphasize how novel strategies, such as dual immune checkpoint inhibition and immune checkpoint inhibition combined with antiangiogenic agents, are beginning to overcome the entrenched resistance within the TME. Looking forward, we discuss the challenges and future directions in HCC treatment, focusing on innovative therapeutic targets, cutting-edge preclinical models, and the potential of integrating spatial multi-omics with digital analytics to help achieve durable responses.

Through this overview, we aim to provide a cohesive narrative that connects epidemiological shifts, mechanistic insights, and clinical breakthroughs, ultimately charting a path toward transforming HCC from a historically intractable malignancy into a disease with more manageable outcomes.

## 2. Global Epidemiological Landscape

Regions with endemic hepatitis B and C virus (HBV and HCV, respectively) infections continue to bear the highest burden of HCC [3]. However, the epidemiological profile is shifting in Western countries (Table 1). In the United States, for instance, metabolic syndrome and its associated metabolic-associated fatty liver disease (MAFLD), metabolic-associated steatohepatitis (MASH), and alcohol-related liver disease have overtaken viral etiologies. Recent improved outcomes in the U.S., with a 5-year survival rate of 21.3%-reflect advances in surveillance, early detection, and multi-disciplinary treatment strategies [4,5].

This epidemiological shift underscores a broader global trend, where rising rates of obesity and type 2 diabetes have propelled metabolic dysfunction to the forefront of HCC etiology, both in the West and in the rest of the world. Recent data indicate that up to 40% of HCC cases in the United States and the European Union are linked to metabolic dysfunction, and importantly, many of these cases emerge in patients without significant cirrhosis [6]. Although hepatitis B still dominates as the leading cause of HCC in most Asian countries, the increasing prevalence of metabolic syndrome—exceeding 30% in some regions such as Japan and South Korea—has begun to shift the disease landscape toward MAFLD-related HCC [7,8]. In the Middle East and North Africa, high diabetes prevalence, reaching around 18% in certain countries, has further fueled the rise of MAFLD-driven HCC [9].

Notably, the occurrence of up to half of MAFLD-related HCC cases in non-cirrhotic livers can often lead to delay of diagnosis until advanced stages, when curative-intent treatments are no longer available. By 2030, MAFLD/MASH is projected to become the leading cause of HCC in the United States and Europe [10,11]. Consequently, there is a growing consensus on the need for enhanced surveillance and earlier intervention in patients at risk. Further, public health strategies targeting obesity, diabetes, and excessive alcohol consumption are critical to improve early detection and intervention.

In summary, while viral hepatitis remains a primary cause of HCC in low- and middle-income countries, where antiviral treatments and vaccination programs have begun to reduce the incidence, MAFLD/MASH has emerged as the second most common and fastest-growing etiology worldwide. These epidemiological shifts underscore the urgent need for tailored public health strategies and clinical interventions to address the evolving landscape of HCC [12].

**Table 1 cancers-17-01860-t001:** Epidemiology of Hepatocellular Carcinoma (HCC) in the United States vs. Global Data [1,5,13,14,15,16].

Parameter	United States (US)	Global (WHO Data)
Incidence Rate	Approximately 6–8 cases per 100,000 people annually	Approximately 110–20 cases per 100,000 people globally (varies widely by region)
Annual New Cases	≈12.5 per 100,000 population new cases (projected for 2024)	≈11.2 per 100,000 population new cases globally
Leading Risk Factors	Hepatitis C infection, alcohol-related liver disease, metabolic-associated fatty liver disease (MAFLD)/MASH	Hepatitis B and C infections, aflatoxin exposure, metabolic-associated liver disease, MAFLD
Most Affected Age Group	50–70 years	Varies by region; generally, affects individuals aged 40–70 years
Gender Distribution	Male—Female ≈ 2–3:1	Male—Female ≈ 2–4:1 (higher male predominance in HBV-endemic regions)
Five-Year Survival Rate	~20% for localized HCC, ~11% overall	~18% globally (varies widely by region and stage at diagnosis); <10% in high-burden regions (due to late diagnosis, limited treatment access)
Primary Etiological Agents	Hepatitis C (historically the most common), but MAFLD and metabolic syndrome-related HCC rates are rising	Hepatitis B is the leading cause globally, especially in Asia and Africa; rising cases from MAFLD in Western countries
Mortality Rate	≈9.3 per 100,000 population deaths annually	≈10.2 per 100,000 population deaths globally annually
Regional Hotspots	Highest incidence rates in Southern and Southwestern states	High-incidence regions include East Asia (China), Southeast Asia, sub-Saharan Africa, and parts of the Middle East
Trends	Increasing due to MAFLD and obesity, especially in younger populations, historically driven by hepatitis C	Rising in Western countries due to MAFLD; stabilized or decreasing in regions with effective hepatitis B vaccination
Screening and Early Detection	Routine screening for high-risk individuals (e.g., those with cirrhosis and hepatitis B/C) is increasing	Varies by region; established in some high-risk areas (e.g., Asia); less common in low-resource settings

## 3. Tumor Microenvironment Landscape in HCC

Hepatocellular carcinoma (HCC) arises within the unique immunological context of the liver, an organ naturally inclined to maintain a delicate equilibrium between immune surveillance and tolerance. While this tolerogenic state is essential for limiting inflammation in response to dietary and microbial antigens, it can become a liability when exploited by malignant cells to evade immune destruction.

HCC typically develops through a stepwise progression initiated by chronic liver injury and sustained inflammation, leading to fibrosis and malignant transformation [17]. Persistent activation of hepatic stellate cells (HSCs) promotes excessive extracellular matrix (ECM) deposition and fibrotic remodeling. This fibrogenic response, combined with cytokine signaling and immune cell infiltration, fosters a pro-tumorigenic microenvironment that supports HCC progression (Figure 1) [18,19,20].

The liver’s specialized sinusoidal architecture, featuring fenestrated endothelial cells and minimal basement membrane, facilitates continuous interaction between hepatocytes, endothelial cells, immune cells, and gut-derived antigens. In chronic liver disease, this homeostatic balance is disrupted. Hallmarks such as hypoxia and fibrosis drive stabilization of HIF-1α, which induces proangiogenic signaling and the formation of disorganized, leaky vasculature. These vascular abnormalities impair immune cell infiltration and reinforce immunosuppressive signaling [21,22]. Simultaneously, stromal activation contributes to matrix remodeling, immune exclusion, and therapeutic resistance through dense ECM deposition and paracrine signaling [23,24].

The tumor microenvironment (TME) in HCC is thus shaped by a complex interplay of immune, stromal, and vascular components that collectively drive tumor progression and limit therapeutic efficacy. Among these, tumor-associated macrophages (TAMs), cancer-associated fibroblasts (CAFs), regulatory T-cells (Tregs), and tertiary lymphoid structures (TLS) represent key cellular populations that orchestrate immune suppression, matrix dynamics, and spatial immune patterning (Table 2). The following sections explore the distinct yet interconnected functions of these cells and their roles in shaping the immunosuppressive, fibrotic, and angiogenic landscape of liver cancer.

In HCC tumors, tumor-associated macrophages (TAMs) predominantly exhibit an M2-like phenotype, secreting vascular endothelial growth factor (VEGF), transforming growth factor-beta (TGF-β), and interleukin-10 (IL-10) to suppress cytotoxic T-cell activity and to facilitate pathologic angiogenesis. Several linked mechanisms reinforce these tumor-promoting behaviors. First, the cleavage of FNDC5 into irisin activates peroxisome proliferator-activated receptor gamma (PPARγ) in TAMs, helping maintain their M2-like state [25]. Second, tumor-derived CCL16 recruits CCR1/CCR5+ monocytes, which differentiate into immunosuppressive TAMs [26]. Finally, the upregulation of matrix metalloproteinase 21 (MMP21) in HCC cells triggers the production of colony-stimulating factor 1 (CSF-1) and fibroblast growth factor 1 (FGF-1), creating a chemotactic gradient for TAM infiltration [27]. Together, these pathways form a self-reinforcing loop that protects tumors from immune surveillance while driving metastatic spread. Beyond their immunosuppressive role, TAMs contribute to ECM remodeling and fibrosis through the secretion of TGF-β and matrix metalloproteinases, linking immune evasion to physical barriers that hinder T-cell infiltration, as noted in various cancers [28,29].

Cancer-associated fibroblasts (CAFs) play a pivotal role in remodeling HCC’s fibrotic tumor microenvironment (TME), both physically and biochemically impeding immune cell infiltration. Activated by TGF-β and PDGF, CAFs secrete dense extracellular matrix (ECM) components such as collagen and periostin (POSTN), effectively blocking cytotoxic T-cells [24]. They also promote therapeutic resistance via paracrine signaling. For instance, CAF-derived exosomes carry oncogenic microRNAs (e.g., miR-1228-3p, miR-20a-5p) that activate the PI3K/AKT and Wnt/β-catenin pathways in HCC cells, reducing sorafenib sensitivity [30,31]. Similarly, secreted phosphoprotein 1 (SPP1) from CAFs diminishes the effectiveness of tyrosine kinase inhibitors (TKIs) by upregulating folate receptor alpha (FOLR1) in tumor cells [32,33]. Furthermore, emerging spatial metabolomics research has identified specialized ECM metabolic niches that actively drive tumor progression and shape therapy response [34,35,36,37,38,39]. CAFs also amplify proangiogenic signaling in response to hypoxia, creating a fibrotic niche that enhances resistance to therapy and supports tumor expansion [23,40].

Regulatory T-cells (Tregs) are key mediators of the immunosuppressive microenvironment in HCC. Enriched particularly in HBV-associated cases, Tregs contribute to immune evasion and tumor progression [41]. They are defined by CD4, CD25, and Foxp3 expression and exert suppressive effects through multiple mechanisms. Tregs inhibit cytotoxic T-cells via inhibitory cytokines (IL-10, IL-35), contact-dependent suppression (CTLA-4, PD-1), and metabolic disruption, such as IL-2 consumption and adenosine production [42]. They are recruited to the tumor through CCR4/CCR6-mediated chemotaxis in response to CCL22 and CCL20, with transcription factors like GATA4 and SOX12 upregulating these ligands [43,44]. In HBV-related HCC, Tregs often display a tissue-resident phenotype and induce apoptosis in effector T-cells via FASLG–FAS interactions [42,45,46].

Tumor-infiltrating Tregs also exhibit enhanced survival and suppressive activity due to epigenetic remodeling [42,45,47]. Clinically, high Treg density correlates with poor prognosis and resistance to therapies, including immune checkpoint inhibitors and sorafenib [41,46,48,49]. Targeting Treg pathways, including CCR4 or CD177, represents a promising strategy to enhance antitumor immunity [44,50,51,52].

While Tregs exemplify cellular mechanisms of immune suppression, the structural organization of immune cells within the tumor microenvironment, such as the formation of tertiary lymphoid structures (TLS), adds another layer of complexity in shaping immune responses and prognosis in HCC [53,54,55]. TLS are ectopic lymphoid aggregates found in some HCCs containing CD8+ T cells, B cells, and LAMP3+ dendritic cells. However, the presence of TLS alone does not consistently predict therapeutic response. Spatial transcriptomic studies reveal that TLS function is contingent on stromal cues. For example, CCL10+ fibroblasts may collaborate with plasma cells (IGHG1+) to recruit T cells, creating immune-active niches, whereas DKK1+ tumor cells can inhibit CCL19+ lymphoid cell infiltration [56,57]. Thus, TLS in HCC correlates with immune therapy responses, but such responses are still dependent on the broader TME context. As the tumor expands, increased metabolic demands and disorganized vasculature lead to localized hypoxia. Hypoxia stabilizes hypoxia-inducible factor 1-alpha (HIF-1α), which upregulates VEGF and other angiogenic factors to promote abnormal vessel formation. These malformed vessels limit immune cell access while further enhancing hypoxia, creating a self-sustaining cycle of immunosuppression and resistance [58,59,60,61].

### Emerging Therapeutic Implications in Remodeling HCC TME

The therapeutic landscape for HCC is rapidly evolving to address the multifaceted challenges posed by its microenvironment. One promising avenue lies in targeting cancer-associated fibroblasts (CAFs). By inhibiting key signaling molecules such as SPP1 and CXCL12 or even by interfering with exosome release, researchers hope to disrupt the paracrine communication that fosters antiangiogenic TKI resistance and promotes tumor survival [32,62,63,64,65]. The reprogramming of tumor-associated macrophages (TAMs) to a more M1-like state is another compelling area of research. Recent studies suggest that modulating pathways like FNDC5/irisin or chemokine receptors such as CCR1/CCR5 could shift TAMs away from their pro-tumorigenic M2-like state toward a phenotype more conducive to antitumor immunity [25,66,67].

Another critical frontier involves the selective depletion or functional modulation of Tregs, particularly by targeting the CCL22-CCR4 or FASLG-FAS signaling axes [44,50,68,69]. These approaches aim to rejuvenate CD8^+^ T cell responses, which are often exhausted by multiple factors in the HCC TME. Recognizing the unique immunological niche of the liver, researchers are increasingly integrating spatial omics and patient-derived models to decipher context-specific interactions within the TME. This comprehensive strategy not only underscores the need for multi-targeted interventions but also offers a roadmap for exploiting the liver’s dual capacity for immune tolerance and surveillance to ultimately enhance therapeutic efficacy in HCC.

## 4. Angiogenesis Signaling in HCC

Angiogenesis in HCC is driven by a complex interplay of cytokines, growth factors, and hypoxia-induced signaling that promotes abnormal vascularization and immune evasion [70]. A hallmark of the HCC TME is hypoxia, which stabilizes HIFs, particularly HIF-1α—a master transcription factor that upregulates key proangiogenic mediators such as vascular endothelial growth factor (VEGF), fibroblast growth factor (FGF), and platelet-derived growth factor (PDGF) [61,71,72,73,74,75,76]. Overexpression of these molecules drives endothelial proliferation and disorganized, leaky blood vessel formation, creating nutrient-rich but immunosuppressive niches (Figure 2) [21,73,75,76]. These malformed vessels exacerbate hypoxia and contribute to a self-perpetuating cycle of tumor progression. Excess VEGF, for example, inhibits dendritic cell maturation and recruits myeloid-derived suppressor cells (MDSCs), enhancing immune evasion [77,78,79].

While HIF-1α is the most extensively studied isoform, growing evidence highlights the role of HIF-2α in regulating angiogenesis and tumor progression in specific HCC subtypes [80,81,82,83]. Preclinical studies have demonstrated that HIF-2α inhibitors, such as PT2385, effectively suppress angiogenic pathways and inhibit tumor progression in HCC models [84,85]. Building upon these findings, clinical investigations are underway to evaluate the efficacy of HIF-2α inhibitors in HCC treatment. Additionally, isoform-specific HIF-2α antisense oligonucleotides (ASOs) have been investigated experimentally in HCC mouse models. These ASOs effectively downregulated HIF-2α expression; however, they did not significantly impact tumor numbers and, in some cases, even aggravated liver fibrosis, highlighting the complexity of therapeutically targeting HIF-2α in HCC [86]. Notably, a Phase I/II clinical trial (NCT04976634) is assessing the safety and efficacy of Belzutifan (MK-6482), a selective HIF-2α inhibitor [46], in combination with Lenvatinib, a tyrosine kinase inhibitor, for patients with advanced HCC. This study aims to clarify the potential benefits and limitations of HIF-2α inhibitors as adjunct therapies, particularly in HCC subtypes characterized by elevated HIF-2α expression. These clinical efforts reflect a growing interest in targeting HIF-2α as a therapeutic strategy to improve outcomes for patients with advanced HCC.

Angiogenic signaling in HCC arises from both local (paracrine) and systemic (endocrine) sources [75,87]. Within the TME, tumor cells and stromal components, including CAFs and TAMs, secret VEGF, IL-6, IL-8, and matrix metalloproteinases promote vessel formation and remodeling [88,89,90,91,92,93]. CAF-derived PDGF recruits pericytes to stabilize nascent vessels; however, this often results in increased vascular permeability and metastatic potential [88,90,91,94]. Importantly, endothelial cells can also emit “angiocrine” regulatory signals to other stromal or even cancer cells in the TME [95]. Deeper insights into this additional layer of function are relatively underexplored but may lead to additional functional vascular targets in the HCC TME.

Systemically, endocrine factors also sustain angiogenesis. Transforming growth factor-beta (TGF-β) induces endothelial-to-mesenchymal transition (EMT), generating hybrid endothelial cells that facilitate vessel co-option [96,97,98,99]. Additionally, insulin-like growth factor-1 (IGF-1) promotes endothelial proliferation through PI3K/AKT signaling while downregulating thrombospondin-1 (TSP-1), an antiangiogenic molecule [100,101,102]. Together, these localized and systemic cues create a proangiogenic environment that supports tumor growth, metastasis, and therapy resistance (Figure 1).

Beyond soluble cytokines and growth factors, recent studies have identified extracellular vesicles, particularly exosomes, as critical mediators that integrate angiogenic and immunosuppressive signals within the HCC microenvironment [103,104,105,106]. These vesicles offer a distinct mechanism of intercellular communication, complementing classical paracrine and endocrine pathways. These nano-sized vesicles, secreted by both tumor and stromal cells, contain diverse molecular cargo such as cytokines, microRNAs, and proteins, representing a distinct mechanism of intercellular communication that complements soluble factor signaling (Table 3).

Exosomes have emerged as key drivers of angiogenic reprogramming in HCC by facilitating crosstalk among tumor cells, stromal components, and pre-metastatic niches [107,108]. This contributes to the establishment of proangiogenic and immunosuppressive conditions that favor tumor progression. Notably, exosomal PD-L1 has been proposed as a potential diagnostic biomarker, reflecting the immunosuppressive dynamics of the HCC tumor [109,110,111]. In parallel, exosomal microRNAs have been implicated in forming proangiogenic niches that enhance neovascularization and metastatic potential [112,113,114]. By modulating both vascular and immune pathways, exosomes reinforce the complex interplay between angiogenesis and immune evasion in HCC, underscoring their relevance as both biomarkers and therapeutic targets.

## 5. Clinical Trials and Treatment Approaches Targeting TME and Angiogenesis

The evolution of HCC treatments over the past decade has not only redefined clinical strategies but also underscored the pivotal role of the tumor microenvironment (TME) in dictating therapeutic outcomes (Table 4). Initially, the introduction of sorafenib in 2007 marked a turning point. As a multikinase angiogenesis inhibitor targeting VEGFR, PDGFR, and RAF kinases, sorafenib demonstrated modest efficacy in the SHARP trial by extending overall survival from 7.9 (with placebo) to 10.7 months. However, its modest benefits and significant side effects demonstrated the need for more refined approaches. Subsequently, additional similar angiogenesis inhibition therapies were evaluated. Cabozantinib [115], regorafenib [116], and ramucirumab [117] offer some benefits after sorafenib treatment. Lenvatinib was found to be non-inferior but perhaps somewhat more easily tolerated than sorafenib [118]. These studies showed that the activity and durability of such monotherapies are often limited by the inability to counteract immune resistance mechanisms.

This recognition of the TME as a key modulator of treatment efficacy has led to the integration of immuno-oncology (IO) agents into therapeutic regimens. The pivotal IMbrave150 trial was the first phase 3 trial to demonstrate the potential synergy resulting from combined targeting of angiogenesis signaling and immune checkpoints [119]. The combination of atezolizumab, a PD-L1 inhibitor, with bevacizumab, an anti-VEGF antibody, showed significantly improved radiographic response rate, progression-free survival, and overall survival compared with sorafenib. Hence, normalizing the tumor vasculature can lead to reinvigorated T-cell function. Similarly, the APOLLO trial (presented in 2024 but not yet published) and CARES-310 trial [120] showed that the combination of antiangiogenic TKIs and PD-1 inhibitors can significantly outperform sorafenib, further reinforcing the value of combining IO with targeted therapies. However, other similar co-targeting strategies, including Lenvatinib plus pembrolizumab [121] or cabozantinib plus atezolizumab [122], did not exhibit similar synergy, pointing to the dynamic and nuanced nature of the HCC TME and the possibility of synergistic toxicity mitigating treatment benefits [123].

Simultaneous inhibition of two critical immune checkpoints—PD1/PD-L1 and the cytotoxic T-lymphocyte-associated protein 4 (CTLA4)/CD80-CD86—has also been explored. The HIMALAYA study [124] demonstrated a significant benefit of a single loading dose of tremelimumab combined with durvalumab, with a very durable benefit in a small but significant minority of treated patients [125]. Similarly, the CheckMate-9DW trial (presented in 2024, not yet published) has shown excellent survival benefits from the combination of ipilimumab (CTLA4 inhibitor) and nivolumab (PD-1 inhibitor)—compared with lenvatinib—despite concerns over a significant proportion of patients experiencing immune-related toxicities.

**Table 4 cancers-17-01860-t004:** Major Clinical Trials in Advanced HCC: Immunotherapy and Combination Approaches.

Category	Trial Name	Therapy Combination	Phase	PFS	OS	Key Findings
IO	CheckMate-040 [126]	Nivolumab (PD-1) + Ipilimumab (CTLA-4)	I/II	5.1 months vs. 4.3 months (nivolumab/ ipilimumab plus cabozantinib)	20.2 months vs. 22.1 months for the triplet arm	Doublet demonstrates encouraging antitumor activity and consistent safety profiles for the combination therapies in patients with advanced HCC.
	KEYNOTE-224 [127,128,129]	Pembrolizumab (PD-1)	II	4 months (95% CI, 2–8)	13.9 months	Pembrolizumab monotherapy provides durable antitumor activity, promising overall survival, and has a manageable safety profile in patients with advanced HCC who have not received prior systemic therapy.
	KEYNOTE-240 [130,131,132]	Pembrolizumab (PD-1)	III	3.0 months vs. 2.8 months (placebo) HR 0.72	13.9 months vs. 10.6 months (placebo) HR of 0.78	Pembrolizumab provides clinical benefits in terms of antitumor activity and safety, supporting its use as a second-line therapy in advanced HCC despite not achieving statistical significance in primary endpoints.
Combination	KN046 [133]	KN046 (PD-L1/CTLA-4 bispecific antibody) monotherapy	II	11.0 months (95% CI, 8.2–15.2)	16.4 months (95% CI, 11.20-not estimable), 12-month OS rate of 60.0% (95% CI, 45.9–71.6)	KN046 combined with lenvatinib showed promising efficacy and a manageable safety profile in patients with advanced unresectable or metastatic HCC. The objective response rate (ORR) was 45.5% (95% CI, 31.97–59.45).
	INSPIRE [134]	Atezolizumab (anti-PD-L1) + Bevacizumab (anti-VEGF)	II	4.8 months	12.9 months	The study highlighted the potential of pembrolizumab as a therapeutic option for HCC, particularly in patients who have progressed on or are intolerant to sorafenib.
	HIMALAYA [135,136]	Durvalumab (PD-L1) + Tremelimumab (CTLA-4)	III	3.8 months (STRIDE) vs. 4.3 months (sorafenib)	16.4 months vs. 13.7 months (sorafenib)	These findings support the use of STRIDE as a novel first-line systemic therapy for unresectable HCC, demonstrating sustained long-term survival benefits and a favorable safety profile.
	IMbrave150 [119,137]	Atezolizumab (PD-L1) + Bevacizumab (VEGF)	III	6.8 months vs. 4.3 months (sorafenib) HR 0.59 (95% CI, 0.47 to 0.76)	19.2 months vs. 13.4 months (sorafenib) HR 0.66 (95% CI, 0.52 to 0.85)	Significant improvement in both PFS and OS, establishing a new first-line treatment standard in HCC. The combination therapy also delayed the deterioration of quality of life and functioning compared to sorafenib.
	CARES-310 [120,138]	Camrelizumab (PD-1) + Rivoceranib (TKI)	III	5.6 months vs. 3.7 months (sorafenib) HR 0.52 (95% CI, 0.41–0.65; one-sided *p* < 0.0001)	22.1 months vs. 15.2 months (sorafenib) HR of 0.62 (95% CI, 0.49–0.80; one-sided *p* < 0.0001).	Demonstrated strong efficacy in PFS and OS, supporting IO-TKI combination as a new and effective first-line treatment option for unresectable HCC.
	CheckMate-9DW	Nivolumab (PD-1) + Cabozantinib (TKI)	III	Not Available	Not Available	Detailed results regarding these endpoints are not available in the current medical literature.
	LEAP-002 [121,135,139]	Pembrolizumab (PD-1) + Lenvatinib (TKI)	III	8.2 months vs. 8.0 months (lenvatinib plus placebo) HR 0.87 (95% CI, 0.73–1.02)	21.2 months vs. 19.0 months (lenvatinib plus placebo) HR 0.84 (95% CI, 0.71–1.00)	While the combination did not significantly improve survival outcomes, it highlighted the potential activity of lenvatinib plus pembrolizumab in advanced HCC with an objective response rate (ORR) of 26.3% for lenvatinib plus pembrolizumab versus 17.5% for lenvatinib plus placebo.

## 6. Spatial Omics and Biomarker Discovery for Precision Medicine in HCC

In addition to traditional approaches such as bulk RNA sequencing, single-cell RNA sequencing, and flow cytometry, emerging spatial profiling methods are emerging as powerful tools for understanding the TME [140]. Techniques spanning single-color immunohistochemistry (IHC), multiplex IHC (mIHC), imaging mass cytometry (IMC), CODEX, and spatial transcriptomics allow researchers to visualize the physical arrangement of interacting cellular subsets and their association with tumor progression, therapeutic resistance, and immune evasion [141]. These spatial insights have informed mechanistic studies and guided the discovery of patient-specific biomarkers.

Recent studies using spatial profiling have revealed substantial intratumoral heterogeneity (ITH) that impacts prognosis and treatment response [142]. Multi-regional sequencing shows that spatial distance correlates with transcriptomic diversity [143]. A composite ITH scoring system combining spatial and molecular data identified that low-ITH tumors express higher levels of CD40 and PD-L1 and are enriched in activated memory T-cells and natural killer (NK) cells, suggesting greater responsiveness to immune checkpoint blockade. In contrast, high-ITH tumors harbor more exhausted CD4^+^ and CD8^+^ T cells and correlate with worse clinical outcomes [144].

Spatial profiling has illuminated key interactions between tumor cells and stromal components that actively shape the immunosuppressive architecture of the HCC microenvironment. CAFs often co-localize with EpCAM^+^ cancer stem cells (CSCs), promoting tumor cell stemness and aggressive phenotypes through direct contact and paracrine signaling [145]. Simultaneously, SPP1^+^ macrophages accumulate in hypoxic zones near CSC-rich regions, fostering a fibrotic, immunosuppressive environment that limits immune cell infiltration [146]. Together, SPP1^+^ macrophages and POSTN^+^ CAFs form physical and biochemical tumor immune barriers (TIBs) that shield tumor cores from T-cell access and reduce the efficacy of immune checkpoint blockade (ICB) therapies [147]. Notably, the blockade of SPP1 in preclinical models disrupts this barrier, restores immune infiltration, and improves ICB responsiveness [148,149].

Additional immunosuppressive structures involve specific macrophage and fibroblast populations that accumulate at tumor boundaries. FAP^+^ CAFs and DAB2^+^ TAMs co-localize at invasive margins in ICB non-responders, forming another type of stromal barrier that excludes cytotoxic lymphocytes [150]. TREM2^+^ macrophages, frequently located near exhausted CD8^+^ T cells, have been implicated in immune tolerance mechanisms and may serve as therapeutic targets. A combined anti-CSF1R and anti-PD1 blockade has shown synergistic antitumor effects in models with TREM2^+^ myeloid populations [151]. Moreover, spatial clustering of TAMs (TAM-high signatures, or TAMs-HS) around tumor nests correlates with poor T-cell infiltration and diminished responses to immunotherapy [152]. Vimentin-high macrophages, which frequently co-localize with Tregs and exhibit elevated inflammatory signaling, may further define an immunosuppressive macrophage subset with predictive value for ICB resistance [153].

The immune architecture of HCC also reflects developmental reprogramming, resembling the immunological state of the fetal liver. The “onco-fetal” phenotype, marked by POSTN^+^ CAFs, FOLR2^+^ TAMs, and PLVAP^+^ endothelial cells, promotes Treg recruitment and immunological tolerance via conserved embryonic signaling programs [154,155]. This developmental mimicry contributes to immune evasion and therapeutic resistance. In metabolic subtypes such as steatotic HCC, the immune landscape is similarly characterized by T-cell exhaustion, infiltration of M2-like macrophages, and dense stromal content, yet paradoxically, these tumors express high levels of PD-L1, suggesting they may still benefit from ICB therapy [156].

### 6.1. Spatial Prognostic and Predictive Biomarkers

Several prognostic models incorporate the quantity and spatial distribution of immune cells, including CD8^+^ T cells [157] and combined T and B cell signatures [158], to predict patient responses to immunotherapies and combination treatments. These spatial metrics serve as powerful biomarkers for patient stratification and outcome prediction.

IMC-based studies comparing proteomic landscapes of primary and recurrent HCCs identified immune features associated with early recurrence. PD-L1+ CD103+ dendritic cells (DCs) were linked to putative interactions between cancer cells undergoing EMT, Tregs, exhausted -cells (Tex), and M2 macrophages, fostering immune evasion [159]. In minimal residual disease (MRD) states, M2-like PDL1+ macrophages interact with CSCs via TGFβ1, contributing to immunosuppression and recurrence. Dual blockade of PD-L1 and TGF-β in animal models restored antitumor activity and eradicated residual CSCs [160].

In addition to individual immune cell subsets, spatially organized immune cell micro-niches can form both at baseline and in response to immunotherapy. For instance, patients who responded to the combination of cabozantinib and nivolumab exhibited higher densities of tertiary lymphoid aggregates (TLAs), along with increased infiltration of CD3^+^/CD8^+^ T cells and CD20^+^ B cells [161]. These structural immune features were accompanied by upregulation of immune-related genes such as CCL19, CXCL14, IGHM, and CXCL6, indicating an active immune environment [162]. Moreover, TLS density at the time of surgery following neoadjuvant treatment with this combination therapy was positively associated with relapse-free survival [53].

### 6.2. Emerging Biomarkers for Dynamic TME Monitoring

Noninvasive biomarkers such as exosomal non-coding RNAs and circulating tumor DNA (ctDNA) are emerging as valuable tools for real-time monitoring of disease progression and immunotherapy (IO) efficacy. These biospecimens encompass analytes originating from malignant cells and diverse non-malignant components within the tumor milieu, including immune infiltrates, stromal fibroblasts, and endothelial elements. Consequently, they provide a dynamic and integrative profile of microenvironmental activity, capturing both tumor-intrinsic alterations and context-dependent responses to therapeutic intervention.

In the INSPIRE trial, ctDNA clearance strongly correlated with durable responses to immune checkpoint inhibitors (ICIs) in patients with castration-resistant prostate cancer. In contrast, in HCC, lower pre-treatment ctDNA levels were linked to a higher likelihood of radiographic complete response [41]. Additionally, multi-omics platforms like the TIMER-HCC score integrate genomic, immune, and stromal features to stratify patients by the predicted response to targeted or immunotherapies [163,164]. This score combines transcriptomic, proteomic, and single-cell RNA sequencing data to map the immune landscape and molecular subtypes of HCC, enabling the identification of biomarkers and immune infiltration patterns associated with therapeutic outcomes [165].

Emerging tools such as spatial technologies and noninvasive biomarkers are transforming our ability to dissect the complex biology of HCC. When combined with artificial intelligence (AI)-driven analytical frameworks, these technologies can uncover hidden patterns within high-dimensional data, offering new insights into tumor heterogeneity, immune evasion, and microenvironmental dynamics. This integrative approach supports the development of more precise and individualized therapies, with the potential to significantly improve outcomes in this challenging malignancy.

## 7. Challenges and Future Directions

Hepatocellular carcinoma remains a formidable challenge, in large part due to its intrinsic cellular heterogeneity and diverse mechanisms of resistance to TME-directed therapies. Hypoxic niches within the tumor microenvironment foster immunosuppression, promote angiogenesis, and shield malignant cells from immune surveillance and attack. These distinct regions, characterized by oxygen deprivation and metabolic stress, can compromise the effectiveness of systemic therapies, including immune checkpoint inhibitors and antiangiogenic agents. Moreover, the dynamic interplay among stromal, immune, and tumor cells evolves during treatment, creating a complex ecosystem that further complicates therapeutic targeting. Looking ahead, the future of HCC treatment appears increasingly promising as researchers explore novel therapeutic targets and emerging technologies to overcome the complexity of the TME. For instance, targeting specific immune cells, such as CCR8^+^ Tregs or MARCO^+^ macrophages, could help dismantle immunosuppressive networks that allow tumors to evade the immune system [166,167]. At the same time, the inhibition of HIF-2α may offer a direct approach to disrupt the hypoxia-driven resistance that undermines current antiangiogenesis therapies.

At the same time, innovations in experimental models and digital technologies are reshaping both preclinical and clinical research. Artificial intelligence (AI)-driven analyses integrating histopathology with multi-omics data are being developed to identify therapeutic vulnerabilities and optimize combination treatments [168,169]. The convergence of novel therapeutic targets, advanced modeling systems, and computational tools opens the door to more effective strategies that may overcome current treatment limitations. For instance, pairing immune checkpoint inhibitors with agents that normalize tumor vasculature, such as bevacizumab, may improve immune cell infiltration while reducing hypoxia-related resistance. In addition, a dual-targeting strategy that addresses both immune evasion and hypoxia pathways could work synergistically to restore antitumor immunity in resistant tumor regions. Together, these emerging approaches offer promise for precision medicine in hepatocellular carcinoma, potentially transforming this challenging disease into a more manageable condition.

The future of HCC therapy lies in spatially informed, multi-target approaches that address the complexity of the TME. Strategies such as dual checkpoint blockade (e.g., anti-PD-1 + anti-CTLA-4) coupled with hypoxia-targeting agents may overcome spatially driven resistance. Meanwhile, liquid biopsy biomarkers, including circulating tumor DNA (ctDNA) and exosomal RNAs, offer noninvasive tools for real-time monitoring of treatment responses and disease evolution. Artificial intelligence-based analytics and multiomics platforms, such as the TIMER HCC score, enable more precise patient stratification and identification of therapeutic vulnerabilities. By integrating these technological and biological advances, the field is moving closer to transforming HCC from a deadly disease into a controllable and manageable chronic condition.

## Figures and Tables

**Figure 1 cancers-17-01860-f001:**
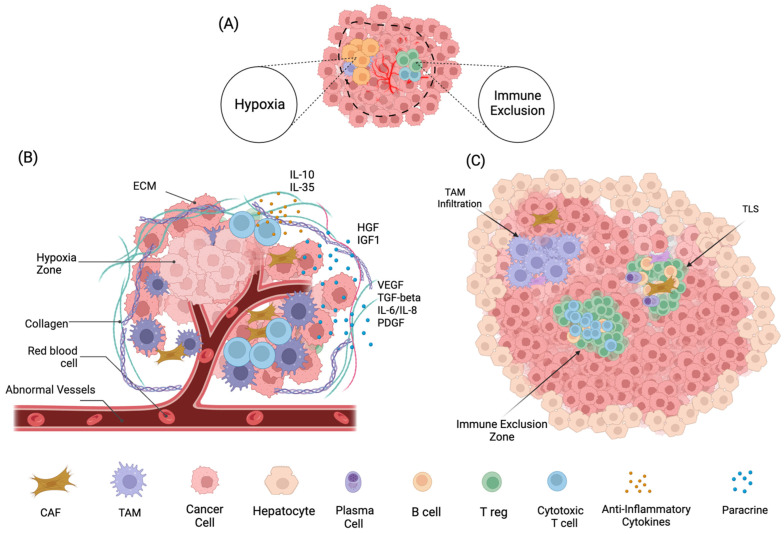
Spatial and Cellular Heterogeneity of the Tumor Microenvironment in Hepatocellular Carcinoma. This illustration highlights key structural and cellular components of the hepatocellular carcinoma (HCC) tumor microenvironment (TME) across different spatial contexts. (A) Schematic overview of the tumor ecosystem depicting disorganized vasculature, abundant stromal fibroblasts, and diverse immune cell infiltration. This cross-sectional illustration of an HCC tumor highlights the complex interactions among tumor cells, immune populations, fibroblasts, abnormal blood vessels, and the extracellular matrix (ECM), which collectively shape the immune microenvironment, drive tumor progression and present opportunities for therapeutic targeting. (B) Abnormal and leaky vasculature contributes to hypoperfusion and hypoxia, impairing immune cell function and promoting extracellular matrix (ECM) remodeling. The perivascular niche is defined by disorganized vessels, collagen deposition, and infiltration of immune cells—including lymphocytes and myeloid-derived suppressor cells—embedded within the ECM. Additional components such as red blood cells (RBCs), stromal fibroblasts, and tertiary lymphoid structures (TLSs) shape the immunomodulatory landscape and influence tumor progression. (C) TLSs are organized aggregates of immune cells resembling lymph nodes, typically forming at tumor margins. Immune exclusion refers to the restricted infiltration of cytotoxic T-cells and other effector immune cells into the tumor core, often due to fibrotic stroma, abnormal vasculature, or immunosuppressive cells. Within the tumor mass, dense clusters of tumor cells are surrounded by exclusion zones and focal TLSs. These barriers may limit immune cell access and reduce the efficacy of immunotherapies.

**Figure 2 cancers-17-01860-f002:**
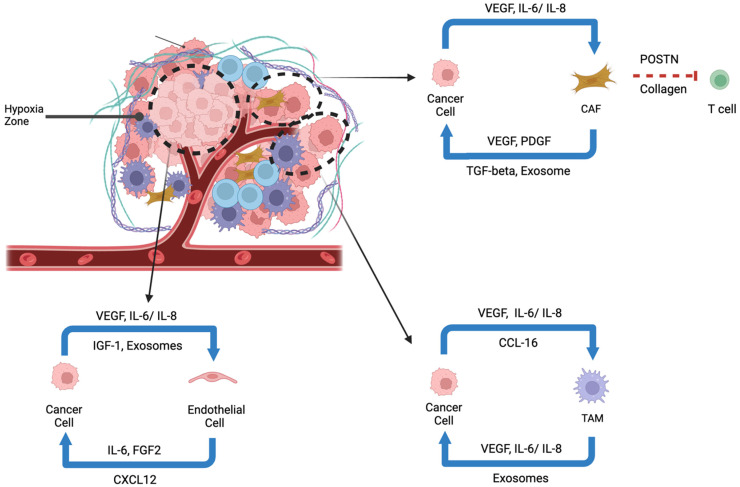
Cytokine and Growth Factor-Mediated Cellular Crosstalk in the Hypoxic Tumor Microenvironment of Hepatocellular Carcinoma. The schematic illustrates cytokine-mediated interactions among cancer cells, endothelial cells, cancer-associated fibroblasts (CAFs), and tumor-associated macrophages (TAMs) in the hypoxic tumor microenvironment of hepatocellular carcinoma (HCC). Cancer cells secrete VEGF, IL-6, and IL-8 to activate endothelial cells (via IGF-1, exosomes) and CAFs (via VEGF, PDGF, TGF-β, exosomes). CAFs contribute to extracellular matrix remodeling through POSTN and collagen, impeding T-cell infiltration. TAMs are recruited via CCL-16 and maintain tumor-promoting loops through VEGF, IL-6, IL-8, and exosomes. These reciprocal interactions support angiogenesis, immune evasion, and tumor progression.

**Table 2 cancers-17-01860-t002:** Key Cell Types in the Tumor Microenvironment (TME) of HCC Compared to Other Solid Tumors.

Cell Type	HCC TME Role	Key Differences in Other Tumors
Cancer-Associated Fibroblasts (CAFs)	Drive fibrosis, immunosuppression, and angiogenesis via TGF-β and HGF; contribute to a dense, fibrotic microenvironment.	Fibrosis is less extensive; CAFs less central in shaping TME.
Tumor-Associated Macrophages (TAMs)	Predominantly M2-polarized; secrete IL-10, VEGF, and TGF-β; promotes immune evasion and angiogenesis.	TAMs are less M2-polarized and not as strongly shaped by liver-induced immune suppression.
Endothelial Cells	Form abnormal, leaky vasculature due to VEGF and hypoxia; enhance hypoxia and limit immune infiltration.	Vascular remodeling is less hypoxia-driven and more structured angiogenesis.
Regulatory T-cells (Tregs)	Abundant due to liver’s immune tolerance; suppress CD8+ T cells and promote immune evasion.	Lower baseline Treg levels and less impact on TME.
CD8+ T Cells	Exhausted or dysfunctional; impaired cytotoxicity due to inflammation and high PD-1 expression.	More functional CD8+ T cells; less exhaustion.
Natural Killer (NK) Cells	Reduced activity due to tolerogenic liver environment and hypoxia limits cytotoxic potential.	NK cells are more active; less suppression by hypoxia or TGF-β.
Myeloid-Derived Suppressor Cells (MDSCs)	Abundant due to inflammation; suppress T-cell activity and enhance immune suppression.	Lower frequency and suppressive capacity of MDSCs.
Extracellular Matrix (ECM)	Densely fibrotic due to liver disease, it forms a physical barrier to immune cells and promotes tumor spread.	ECM remodeling is driven by tumor cells, not pre-existing fibrosis.

**Table 3 cancers-17-01860-t003:** Key Paracrine and Exosomal Mediators in HCC.

	Factor	Source	Mechanism
Paracrine	VEGF	Tumor cells, CAFs	Binds VEGFR2 to induce endothelial proliferation and vascular permeability.
	IL-6/IL-8	TAMs, HCC cells	Activates STAT3 in endothelial cells, enhancing survival and MMP expression.
	PDGF	CAFs	Recruits pericytes to stabilize nascent vessels, albeit abnormally, fostering leaky vasculature that facilitates metastasis.
	HGF	Stromal cells, plasma	Triggers c-MET signaling to promote vascular mimicry and metastasis.
	TGF-β	CAFs, Tregs	Induces EndMT, enabling vessel co-option and immune evasion.
	IGF-1	Tumor cells	promotes endothelial proliferation via PI3K/AKT signaling, while suppressing antiangiogenic thrombospondin-1 (TSP-1).
Exosomal Cargo	PD-L1	Tumor cells	Suppresses T-cell activation to promote immune evasion. Potential biomarker for immunosuppression.
	miR-210	Tumor cells	Suppresses Ephrin-A3, destabilizing endothelial junctions to aid intravasation. Associated with increased vascularization.
	miR-122	Tumor cells	Regulates cell proliferation and modulates drug sensitivity. Investigated as a diagnostic/therapeutic response marker.
	miR-21	Tumor cells, TAMs	Enhances cell proliferation and invasion. Considered a prognostic marker for HCC progression.

Factor

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
