# Peer review of "Bridging Immune Evasion and Vascular Dynamics for Novel Therapeutic Frontiers in Hepatocellular Carcinoma"

_cancers, 2025, doi:10.3390/cancers17111860_

Round 1
Reviewer 1 Report
Comments and Suggestions for Authors
The submitted Manuscript by Wu et al., titled "Bridging Immune Evasion and Vascular Dynamics for Novel Therapeutic Frontiers in Hepatocellular Carcinoma" is important summary of current epidemiological trends of HCC, diverse HCC TME and evolution of immunotherapeutic strategies. There is a tremendous need for better therapeutic options against this disease, hence here submitted manuscript is clinically relevant. Additionally, it is well written, logically presented and supported by appropriate references. With minor modifications listed below, the manuscript is in appropriate form for publication in Cancers.
Specific comments:
- In the current form of the Manuscript, Tables 2 and 3 are presented before the Table 1.
- Mediators of HCC are listed as paracrine, endocrine, and exosomal. In the Table 2B, HGF (secreted by stromal cells and plasma (source in plasma should be additionally explained)), TGF-β (secreted by CAFs and Tregs) and IGF-1 (secreted by tumor cells) are listed as endocrine. Since they are all sourced by tumor components, they should be preferentially listed as paracrine not endocrine.
- Additional schematics (Figures) would be helpful for better understanding and cleaner presentation.
Author Response
Reviewer 1
The submitted Manuscript by Wu et al., titled "Bridging Immune Evasion and Vascular Dynamics for Novel
Therapeutic Frontiers in Hepatocellular Carcinoma" is important summary of current epidemiological trends of
HCC, diverse HCC TME and evolution of immunotherapeutic strategies. There is a tremendous need for better
therapeutic options against this disease, hence here submitted manuscript is clinically relevant. Additionally, it is
well written, logically presented and supported by appropriate references. With minor modifications listed below,
the manuscript is in appropriate form for publication in Cancers.
Response 1.0: Thank you for the summary.
Specific comments:
1. In the current form of the Manuscript, Tables 2 and 3 are presented before the Table 1.
Response 1.1: Thank you for catching this. We have adjusted the formatting of the revised manuscript
accordingly, with Table 1 now appearing on page 3, Table 2 (formerly “Table 2A”) on page 6, Table 3 (formerly
“Table 2B” on page 9, and Table 4 (formerly “Table 3”) on pages 10-11.
2. Mediators of HCC are listed as paracrine, endocrine, and exosomal. In the Table 2B, HGF (secreted by
stromal cells and plasma (source in plasma should be additionally explained)), TGF-β (secreted by CAFs
and Tregs) and IGF-1 (secreted by tumor cells) are listed as endocrine. Since they are all sourced by
tumor components, they should be preferentially listed as paracrine not endocrine.
Response 1.2: Thank you for helping us to clarify our descriptors. We have modified the classification to include
only paracrine and exosomal mediators in the revised Table 3 on page 9.
3. Additional schematics (Figures) would be helpful for better understanding and cleaner presentation.
Response 1.3: Thank you for the feedback. We have now added a new Figure 2 (on page 8) which we believe
more specifically illustrates specific cellular crosstalk interactions.
Reviewer 2 Report
Comments and Suggestions for Authors
This is a review paper focusing on the recent knowledge on the global epidemiological trends, the cellular and molecular landscape of the HCC TME, and the latest clinical advances in immunotherapy and combination treatments. This is a nice manuscript emphasizing the need for multitargeted approaches to synergistically modulate interacting cellular constituents and ultimately improve clinical outcomes. Description of Global Epidemiological and Tumor Microenvironment Landscapes in HCC, Angiogenesis Signaling, Clinical Trials and Treatment Approaches, Spatial Omics and Biomarker Discovery, as well as Challenges and Future Directions -are included in 7 chapters. The manuscript is of interest, written in good English and could be accepted for publication after minor revision.
Comments
- Please check the Cancers Tables style. I think Tables usually do not have explanations at the bottom.
- Tables are sometimes difficult to read, as the lines are moving up and down. Please check the horizontal alignment in the Tables.
- Please align the text in Figure 1 (cell names and pictures).
- Please check all the font sizes and make the font same in style and size (“Angiogenesis Signaling in HCC” section).
Author Response
Reviewer 2
Comments and Suggestions for Authors
This is a review paper focusing on the recent knowledge on the global epidemiological trends, the cellular and
molecular landscape of the HCC TME, and the latest clinical advances in immunotherapy and combination
treatments. This is a nice manuscript emphasizing the need for multitargeted approaches to synergistically
modulate interacting cellular constituents and ultimately improve clinical outcomes. Description of Global
Epidemiological and Tumor Microenvironment Landscapes in HCC, Angiogenesis Signaling, Clinical Trials and
Treatment Approaches, Spatial Omics and Biomarker Discovery, as well as Challenges and Future Directions -
are included in 7 chapters. The manuscript is of interest, written in good English and could be accepted for
publication after minor revision.
Response 2.0: Thank you for the summary.
Comments
1. Please check the Cancers Tables style. I think Tables usually do not have explanations at the bottom.
Response 2.1: Thank you for clarification. After consultation with the editorial staff, we have removed the Table
explanations as suggested.
2. Tables are sometimes difficult to read, as the lines are moving up and down. Please check the horizontal
alignment in the Tables.
Response 2.2: After discussion with the Cancers editorial staff, we have modified the table formatting as follows:
1) Left-justified table columns with text of length greater than a few words. 2) slightly modified the line breaks to
increase blank spacing between columns’ text.
3. Please align the text in Figure 1 (cell names and pictures).
Response 2.3: Thanks for this feedback. We have aligned the center of each text descriptor to the middle of the
illustration for each cell type in the revised Figure 1.
4. Please check all the font sizes and make the font the same in style and size (“Angiogenesis Signaling in
HCC” section).
Response 2.4: Thank you for the feedback. We have modified this section and ensured that the font sizes in
this section are consistent with the remainder of the revised manuscript.